# Fiber Bragg Grating Sensors for Reinforcing Bar Slippage Detection and Bond-Slip Gradient Characterization

**DOI:** 10.3390/s22228866

**Published:** 2022-11-16

**Authors:** Luis Pereira, Esequiel Mesquita, Nélia Alberto, José Melo, Carlos Marques, Paulo Antunes, Paulo S. André, Humberto Varum

**Affiliations:** 1I3N & Department of Physics of the University of Aveiro, Campus Universitário de Santiago, 3810-193 Aveiro, Portugal; 2Laboratory of Rehabilitation and Buildings Durability, Campus Russas, Federal University of Ceara, 62900-000 Russas, Ceará, Brazil; 3Instituto de Telecomunicações and University of Aveiro, Campus Universitário de Santiago, 3810-193 Aveiro, Portugal; 4CONSTRUCT-LESE, Faculty of Engineering of the University of Porto, Department of Civil Engineering, Structural Division, 4200-465 Porto, Portugal; 5Instituto de Telecomunicações and Department of Electrical and Computer Engineering, Instituto Superior Técnico, University of Lisbon, 1049-001 Lisboa, Portugal

**Keywords:** bond-slip, Fiber Bragg Gratings, pull-out testing, reinforcing concrete, structural health monitoring

## Abstract

The detection of bond-slip between the reinforcing bar (RB) and concrete is of great importance to ensure the safety of reinforced concrete (RC) structures. The techniques to monitor the connection between the RB and concrete are in constant development, with special focus on the ones with straightforward operation and simple non-intrusive implementation. In this work, a simple configuration is developed using 10 optical fiber sensors, allowing different sections of the same RC structure to be monitored. Since the RB may suffer different strains along its length, the location of the sensors is critical to provide an early warning about any displacement. Bragg gratings were inscribed in both silica and polymer optical fibers and these devices worked as displacement sensors by monitoring the strain variations on the fibers. The results showed that these sensors can be easily implemented in a civil construction environment, and due to the small dimensions, they can be a non-intrusive technique when multiple sensors are implemented in the same RC structure.

## 1. Introduction

Being the most popular construction material in civil engineering, concrete is widely used in all types of constructions in combination with other materials, such as steel reinforcing bars (RBs), to obtain better structural behavior and to take advantage of the material strengths. To reduce errors and consequently increase the quality and maintenance of reinforced concrete (RC) structures, monitoring the construction phase and the lifespan of the structure is essential. The bond-slip between the RB and concrete may occur during the concrete curing process, since the steel bar suffers physical changes caused by the chemical and thermal transformations in the concrete. The RB may suffer different strains throughout this process [1], affecting the adherence zone between the RB and the concrete. However, it is during the lifespan of a construction that it is more important to monitor the bond-slip. The combination of external forces (like earthquakes and hurricanes) with constant exposure to climate changes and thermal contraction/expansion creates several forces and tensions in the structure which affect the steel–concrete adhesion and could lead to damage or even to the collapse of the structure [2].

The integrity of the adherence zone is crucial for safety maintenance in RC structures and it can be analyzed by crack analysis, a procedure that can be used to diagnose the bond state of the concrete with the embedded bars [3]. Besides other factors like age, material, geometry and loading, the crack evolution is affected by the tension distribution in the surrounding area between RB and concrete. To analyze the steel–concrete adhesion through the bond-slip evolution, a pull-out test is often performed, which shows valuable information including the maximum limit of the bond strength [3,4,5,6].

In order to detect any displacement between the steel RB and the concrete during the lifespan of a RC structure, a long-term monitoring is necessary. To do so, the sensors must be small or non-intrusive, durable in an adverse environment with concrete (corrosive and with strong tensions during the curing process), low cost and easy to install. One way to monitor the bond-slip is using optical sensors, since they allow easy implementation and reliable results. The employment of optical fiber sensors in concrete and RC structure monitoring applications began in 1980s [7], and since then multiple types of sensing devices have been investigated and developed. Among them, fiber Bragg grating (FBG) sensors have been shown to possess immense potential and suitability in this type of applications, as a cost effective and relative simpler method to measure bond-slip without interfering with RC structural properties and characteristics or even the adherence area between the RB and the concrete. Davis et al. [8] investigated a distributed FBG sensing system focused to monitor the strain at multiple locations through RC beams and decks, while they were tested to failure. Kenel et al. [9] installed FBG sensors along the RBs embedded in RC beams, which were subjected to bending. These sensors could measure large amounts of strain and strain gradients with high precision, without compromising the bond properties. Wang et al. [10] used embedded FBG sensors into fiber-reinforced-polymer bars to create smart bars for near-surface mounted method applications. The smart bars were used for structural strengthening and strain monitoring, due to their mechanical proprieties and embedded sensors, respectively. Ho et al. [11] reported a method to measure the strain of a prestressing tendon during bond-slip by installing FBG sensors directly onto the tendons within a prestressed concrete girder to provide local perspective of the bond-slip as the girder was tested to failure. The local strains measured by this sensing configuration permitted the monitoring of the entire bond-slip process, from the moment the slippage took place until complete bond failure. Mesquita et al. [12] developed two sensor prototypes based on silica and polymer FBGs, with thermal compensation, which were attached to the RB inside a concrete block specimen to monitor the slippage displacement during a pull-out test. The obtained results showed that during the pull-out test the FBG-based prototypes presented better signal-to-noise ratio than a linear variable differential transformer (LVDT), which demonstrates the viability of the optical sensors during the early detection of reinforcement slippage in RC structures. Jayawickrema et al. [13] investigated the application of an enclosed FBG sensor embedded in a concrete beam for monitoring structural integrity. It was observed that the internal deformations in concrete structures, created by the flexural loading, can be detected and measured by the FBG sensors. Čápová et al. [14] installed sensors containing FBGs in a concrete bridge by attaching those sensors to the reinforcing steel to monitor the prestressing of the structure in the initial phase and later the overall load of the structures, the impacts of temperature changes and traffic intensity. Kaklauskas et al. [15] performed experimental and numerical studies to analyze the strain distribution on RC short tensile specimens by installing tensor strain gauges and FBG sensors in the RB. Although the recordings from the FBGs were not as accurate as the strain gauges, the study highlighted the advantages of the FBG sensors in this type of application, namely the simple installation process, non-intrusive characteristics, the capability of using FBG sensors in smaller bar dimeters and the possibility of employing larger number of sensors with finer spacing between them. Another approach to evaluate the adherence zone between the reinforcing bar and the concrete, as well as the durability of the structure, is to detect and monitor corrosion, using FBGs to measure strain, temperature and refractive index in the surroundings of the RB [16]. In the case of corrosion monitoring based on strain measurements, the FBG sensors are installed in the reinforcing bar to detect strain variations from the corrosion-induced expansion, as the corrosion products accumulate as result of the corrosion process [17,18,19]. These works that aim to study and monitor strain in concrete structures, as well as the investigation, development and application of other optical fiber sensors in structural health monitoring (SHM), are boosted by the increasing recognition of the potential of this type of sensors and their compelling advantages in comparison with conventional sensors. Features such as immunity to electromagnetic interference, the absence of electricity at the measuring point, the possibility of multiplexing various sensors in a single optical fiber, small dimensions and consequently low intrusion, and reliable performance with high resolution are some of the advantages that allow the optical fiber sensors, and particularly FBG sensors, to be implemented in civil engineering [20].

The present work details an experimental configuration to monitor the slippage of the RB in different sections of a RC sample. This configuration is composed of 10 sensors based on FBG, located in five different sections of the RC sample, which allows quasi-distributed based measurements. The capability to monitor the RB displacement in different sections along its length increases the efficiency to detect and locate any collapsed adherence zone between the RB and the concrete and allows for analysis of the bond-slip gradient, providing important information to assess the safety of the structure. The Bragg gratings were inscribed in both silica and polymer optical fibers (POFs) to confirm the performance of both fibers in this configuration, and to explore some of the POF characteristics that show great potential for civil engineering applications, such as high elastic strain limit, high flexibility and high fracture toughness [21]. 

## 2. Materials and Methods

### 2.1. Sensor Description and Manufacturing

To monitor the RC bond-slip, 10 sensors were produced by photo-imprinting a Bragg grating in close proximity to one of the extremities of the optical fibers. Figure 1a shows the details of this sensor. Each one consists of passing the optical fiber with a FBG inside a protective sleeve that allows it to move freely inside. The fiber tip was then attached to the RB and an obstacle, which acts as an anchor and was attached to the concrete (see Figure 1b). Note that the FBG sensor was located between the fiber tip (attached to the RB) and the anchor (attached to the concrete). This way, whenever there is a displacement between the RB and the surrounding concrete, the end of the fixed fiber follows the movement of the bar while the anchor remains fixed to the concrete, increasing the strain exerted on the fiber. The strain variations on the fiber provoked by the displacement of the bar shifts the Bragg’s wavelength of the gratings, and consequently it is possible to monitor and measure that displacement in real time. 

All the FBGs employed in the bond-slip sensors were produced by the phase mask method, which is the simplest and the most common method [22,23]. These sensors were divided in two groups, each corresponding to the type of optical fiber the grating was inscribed to: silica optical fiber Bragg grating (SOFBG) and polymer optical fiber Bragg grating (POFBG).

For the manufacturing of the SOFBGs, the fiber used was a step index single-mode (SM) GF1 model from Thorlabs, with core diameter of 10 μm and cladding diameter of 125 μm. The 10 mm FBGs were inscribed by the phase mask method, using a KrF excimer laser system (Coherent Bragg Star Industrial-LN), which emits at a 248 nm wavelength. The SOFBGs were produced using 15 ns pulse repetition rate of 500 Hz with an energy of 5 mJ during 30 s.

Using the same inscription method, the FBGs were also produced in step index SM poly(methyl methacrylate) (PMMA) POF; for details of the fabrication see [24]. Currently, the majority of the POFs available are based on PMMA, but other polymer materials have gained attention due to their unique capabilities and advantages for sensing applications [25,26,27]. The used SM PMMA POF had a core diameter and an outer diameter of approximately 8 μm and 125 μm, respectively. POFBG manufacturing occurred in several fiber samples, approximately 10 cm in length, by irradiating from above a phase mask placed on top of the POF with the 325 nm UV light from a Helium–Cadmium (HeCd) laser, using an output power of 30 mW. To monitor the gratings production, a butt-coupled connection was made between the PMMA POFs and a fiber connector/angled physical contact (FC/APC) silica fiber pigtail, where index matching gel was added in order to reduce Fresnel reflections, lowering the background noise. The FC/APC pigtail was connected to a 1550 nm 50:50 ration SM silica coupler, in which a broadband light source (from Thorlabs ASE-FL7002-C4 centered at 1560 nm) and an optical spectrum analyzer (OSA) were connected to the opposite arms to monitor the reflective spectral characteristics of the 10 mm POFBGs during the inscription process. After 30 min of inscription time, the POFBG samples were UV-glued, with Norland 78, to the FC/APC pigtails.

Before attaching the FBG sensors in the RB, their response to strain was characterized. Since the produced SOFBGs were physically identical, fabricated in the same optical fiber by the same method and laser system in the 1550 nm spectral region, it was expected that all of them had similar strain sensitivity; therefore, a random SOFBG sample was chosen to study the Bragg grating response to strain. The same occurred with the POFBGs, where a random POFBG sample was chosen to analyze their strain sensitivity. The spectral response of the SOFBG and POFBG to strain variations was monitored with an optical interrogator (Micron Optics, model sm125) and the strain characterization was performed by fixing each fiber between a fixed and a manual translation stage, with 5 μm resolution. In both fibers, the strain was increased up to 500 με, with steps of 50 με. The response of the SOFBG and POFBG samples is presented in Figure 2a,b, respectively.

For the SOFBG sample, the central wavelength variation with the increasing strain showed a sensitivity of 1.210 ± 0.003 pm/με, which is in accordance with the typical value found for commercial GF1 and standard single-mode optical fibers (1.2 pm/με) [12,28]. In the POFBG sample the obtained strain sensitivity was 1.53 ± 0.02 pm/με. This value is relatively close to the one found in [29] (1.46 pm/με), where a similar step index SM PMMA POF with a slightly larger diameter (133 μm) was used and fit in the typical values found for POFs with gratings in the 1500 nm region [30].

### 2.2. Description of the Experimental Setup

The bond-slip characterization was performed in a RC pull-out specimen prepared according to Annex D of the EN 10080 [31], using both SOFBG and POFBG sensors simultaneously. For the experimental setup, a concrete block with dimensions 0.20 m × 0.20 m × 0.30 m with a centralized 16.00 mm diameter RB was manufactured. According to proceedings in EN 12390-3 [32], the mean compressive strength of the concrete cubic sample was about 17.6 MPa. In the experimental test, four SOFBG sensors and six POFBG sensors were employed, glued to the RB, and arranged according to Figure 3. 

Two SOFBG sensors were located at 100.00 mm and 200.00 mm from the surface of the concrete block on the side of the RB was pulled out (right side of the RC sample in Figure 3), while the other two were located at the same distance, but on the opposite side of the RB. The POFBG sensors were located in pairs at 50.00 mm, 150.00 mm and 250.00 mm from the right side of the RC sample, also on opposite sides of the RB. With the sensors attached, the RB was placed in the center of the box (formwork structure) to be embedded in concrete. Figure 4 shows the formwork structure after filling it with concrete.

All the sensors were connected to an optical interrogation system (Micron Optics, model sm125) and during the initial 24 h they were monitored in order to identify possible damages and signal losses provoked during the concreting step and material curing. The sensors were later disconnected from the optical interrogator, and the RC sample went through the concrete cure process in a climate chamber, with relative humidity of 95% and temperature of 20 °C, over 28 days. The pull-out test was performed following the descriptions of the Annex D of the EN 10080 [31], and the details are illustrated in Figure 5. The RC sample was positioned in a centralized form and immobilized by two stiff metallic plates, fixed in the inferior part of the steel frame. Then, the optical sensors were connected to the interrogation system for data acquisition, and the pull-out testing was carried out, with the RB being pulled vertically by a mechanic claw. The vertical servo-actuator moved with a displacement velocity of 0.10 mm/s during the pull-out test.

The response of the optical sensors was then studied by displacing the RB of the RC and monitoring the reflection signal of the Bragg gratings. Since the pull-out of the RB applies strain on the optical sensors, the displacement measures can be obtained knowing the relationship between the strain variation and the Bragg wavelength shift (strain sensitivity) for each FBG sensor. When the optical fiber is subjected to a mechanical perturbation, the strain will be directly related to the applied force (Hooke’s law), as the following equation demonstrates:(1)$$K=\frac{\left|F\right|}{\left|\Delta l\right|}$$
where *K* is the elastic constant and Δl is the length variation of the optical fiber provoked by the external force, *F*. The relation between the optical fiber Young’s modulus (*E_f_*) and *F* is given by:(2)$$F=E_{f}.\;A.\frac{\Delta l}{l}$$
where *A* is the cross-sectional area and *l* is the length of optical fiber. The strain is given by $\frac{\Delta l}{l}$. Using Equations (1) and (2), the elastic constant, *K*, can be rewritten to:(3)$$\left|K\right|=\frac{E_{f}.A}{l}$$

By monitoring the Bragg wavelength spectral position of the reflected signal, the strain applied to each fiber is known, giving the Δl values, which are about the same as the RB displacement values.

## 3. Results and Discussion

The results of the pull-out test are presented in Figure 6.

The FBG sensors were displayed and identified according to Figure 6a. At 642 s after the beginning of the pull-out test, the experimental procedure was suddenly interrupted when the RB broke near the region attached to the mechanical claw. Despite this failure, the data collected by the optical sensors was sufficient to characterize the initial slippage of the RB. Figure 6b shows the force applied to the RB during the pull-out test (Trial 1), and Figure 6c shows the relative displacement yielded by the servo-actuator during the same test. The displacement measured by the FBG sensors is shown in Figure 6d,e, where the sensors placed near of the mechanic claw (external force source) registered higher displacement measures.

The obtained results indicate that the adherence zone between the RB and the surround concrete, where sensors S5 and S10 were placed, was the most affected by the external force. Also, due to the fact of being in the closest zone of the RC sample to the external force source, sensors S5 and S10 were the first to be affected during the pull-out test, which in the real scenario of bar slippage could indicate the location of possible external force sources and the location of the adherence zone section already damaged by that force. The maximum displacement values measured by sensors S5 and S10 were 305.32 µm and 288.93 µm, respectively, just before the interruption of the test. The strain on these sensors started to increase drastically after 80 s since the beginning of the test, at which point the adherence zone started to be affected. The other sensors were also affected, as the other sections of adherence zone of the RC sample where the sensors were located began to fail to the force applied on the RB. The last section of the RC sample to be affected by the pull-out was the zone where sensors S1 and S6 were located, measuring respectively only 2.94 µm and 2.86 µm of relative displacement before the interruption of Trial 1. During the test, optical sensors S3 and S4 failed to get any displacement measures, possibly due to connection issues or fiber damage during the cure process of the concrete. Since the RB was not withdrawn from the concrete block in Trial 1 and therefore the pull-out test was unfinished, a second test (Trial 2) was carried out with the same RC specimen and the obtained results are demonstrated in Figure 7.

In the second pull-out test, which was performed under the same conditions, the force applied to the RB during the experiment is shown in Figure 7a, while the relative displacement yielded by the servo-actuator is shown in Figure 7b. The experimental results from the optical sensors are depicted in Figure 7c,d, indicating once more that the displacement was higher on sensor S5, with a maximum value of 462.66 µm just before pulling out the bar. This sensor was also the first to be affected due to its proximity to the external force source. However, sensor S10 did not measure any displacement, possibly because the fiber detached from the RB after Trial 1. Just like the previous test, the last section of the RC sample to be affected by the pull-out was the zone where sensors S1 and S6 were located, with maximum displacement of 3.70 µm and 8.53 µm, respectively.

Comparing the previous results, the slippage of the RB showed identical behavior in both pull-out tests (Trial 1 and Trial 2), where the optical sensors closer to external force source (S5 and S10) were the first to detect the slippage and eventually were the ones that measured higher displacement during the tests. On the other hand, due to their location on the opposite side of the RC specimen, sensors S1 and S6 detected very small displacements in both tests. The maximum displacement measured by these sensors was 2.94 µm (S1) and 2.86 µm (S6) in Trial 1, and 3.70 µm (S1) and 8.53 µm (S6) in Trial 2 (just before pulling out the bar). These results show that the optical sensors are able to detect the initial slippage, and the obtained data allow the displacement of the RB in different sections to be analyzed when an external force is applied and a profile of the relationship between the adherence strength and the applied external force to be obtained. It is also important to point out that Trial 2 is a continuation of the unfinished Trial 1, thus the initial conditions of the RC specimen are different, especially the ones related to the adherence zone between the RB and the concrete, since an external force was already applied to the bar during Trial 1 and its displacement was detected by the optical sensors. Hence, some optical sensors began to detect the RB displacement much sooner during Trial 2, approximately 20 s after the beginning of the test, just after starting to apply force (at that moment the applied force was 2.18 kN), while during Trial 1 they began to detect approximately 80 s after the beginning of the test (at that moment the applied force was 57.04 kN).

The displacement values measured by the optical sensors at 600 s in Trial 1 and at 200 s in Trial 2 are presented in Figure 8a and Figure 8b, respectively.

The missing data is from optical sensors S3 (15 cm position) and S4 (10 cm position), which failed to get any measures in both Trials, and from sensor S10 (5 cm position), since it did not detect any displacement during Trial 2. The results from Figure 8 demonstrate that different sections of the adherence zone were differently affected by the external force in the RB at those times. It is also demonstrated that the bar suffered different percentages of strain along its length; this means that the closer to the external force source, the greater the strain and consequently the greater the bar slippage. This behavior is typical from the relation between the external force and the bond forces between the bar and the surrounding concrete as demonstrated by [33], showing that the adherence zone section close to the external force is the first to collapse, and after which the following sections collapse successively. The same behavior is demonstrated in Figure 9a,b, where for the same external force (100 kN) in Trial 1 and Trial 2, respectively, the displacement measured by the optical sensors was higher as the sensor location was closer to the mechanical claw, showing that the strain on the bar was higher in that region.

In addition, the strain on the bar decreases almost exponentially along its length, as the distance from the external force source increases. By comparing the results from both tests, the bond-slip measures show great discrepancy, namely in sensor S5, which registered bar displacements of 69 (Trial 1) and 290 µm (Trial 2). As mentioned before, Trial 2 was performed to conclude the unfinished Trial 1, which failed to withdraw the RB from the concrete block, and for that reason the discrepancy between the obtained results can be explained by the differences in the adhesion capability due to the formation of cracks and voids in the adherence zone during the first pull-out test.

## 4. Conclusions

In this work, we showed an effective sensing approach to detect and monitor bond slip in RC structures, using both SOF and POF. In total, 10 optical sensors were used during the pull-out tests (Trial 1 and Trial 2), four SOFBGs and six POFBGs, all attached to the RB. During the entire experiment, both SOFBG and POFBG sensors showed similar bond-slip detection response, monitoring capabilities and limitations. However, these similarities are the result of the nature of the pull-out test. Despite the similar sensing capabilities of the POFBG and SOFBG devices, POFs are much more elastic than silica fiber and have higher strain limits, which allows higher displacement values to be monitored. However, as the entire adherence zone collapsed during Trial 2, the tension accumulated in RB was released and its sudden displacement surpassed the strain limits of the sensors (SOFBGs and POFBGs), hence it was not possible to verify the elastic capabilities of the POFBGs. Nevertheless, this quasi-distributed sensing configuration allowed for monitoring of different sections of the RC sample and evaluating the bar displacement in those sections during the test. Also, the fact of having several sensors at the same sample allowed monitoring to continue during the entire pull-out test, even when it was no longer possible to acquire the optical signal from some of those sensors. This is important in a real scenario, since this approach can still provide an early warning about any slippage between the RB and the concrete, even with only a few sensors operating. In Trial 1, just before the interruption of the pull-out test, the maximum displacement was measured by the sensors located closer to the external force, S5 and S10, and the obtained values were 305.32 µm and 288.93 µm, respectively. During Trial 2, despite the failure of sensor S10 in measuring the movement of the RB, sensor S5 was able to register a maximum displacement of 462.66 µm just before the pull-out of the RB. On the other hand, the last section of the RC sample to be affected by the pull-out was the zone located further from the external force where sensors S1 and S6 were installed, which measured maximum displacements of 3.70 µm and 8.53 µm, respectively, during Trial 2. Also, since Trial 2 is the repetition of the unfinished Trial 1, the initial conditions of the adherence zone between the RB and the concrete were different due to the external force applied to the bar during Trial 1 and consequently the optical sensors started measuring the RB displacement sooner during Trial 2 (at approximately 20 s, while in Trial 1 was at approximately 80 s). This can also explain the discrepancy between the maximum displacement values measured by the sensors in Trial 1 (69 µm) and in Trial 2 (290 µm), when the RB was subjected to the same applied force (100 kN). The results of this work demonstrate that different sections of the adherence zone were differently affected by the external force in the RB during pull-out tests. Since the RB suffered different percentages of strain along its length, the location of the sensors is a key factor when monitoring the RC structures, due to the fact that in a certain moment of time some sections of the RB would displace while others would not. Thus, this configuration has the potential to increase the efficiency and accuracy when monitoring bond-slip in RC structures, allowing the collapsed adherence zone between the RB and the concrete to be located and the bond-slip gradient to be analyzed. Also, despite the promising results, more tests are needed to confirm the viability of this sensing configuration, where the limitations of POF and silica fiber sensors are studied more intensively, the location and space between sensors are analyzed, and the parameters of the pull-out test as well as the RC samples are altered to analyze the sensors performance under different bond-slip conditions.

## Figures and Tables

**Figure 1 sensors-22-08866-f001:**
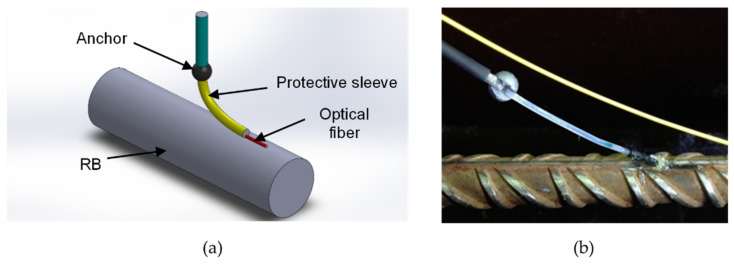
Details of the bond-slip sensor: (**a**) Sketch of the sensor. (**b**) Final aspect of the sensor attached to the RB.

**Figure 2 sensors-22-08866-f002:**
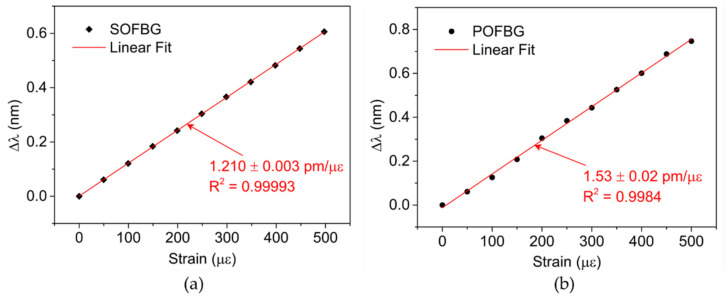
Wavelength response to strain: (**a**) SOFBG. (**b**) POFBG.

**Figure 3 sensors-22-08866-f003:**
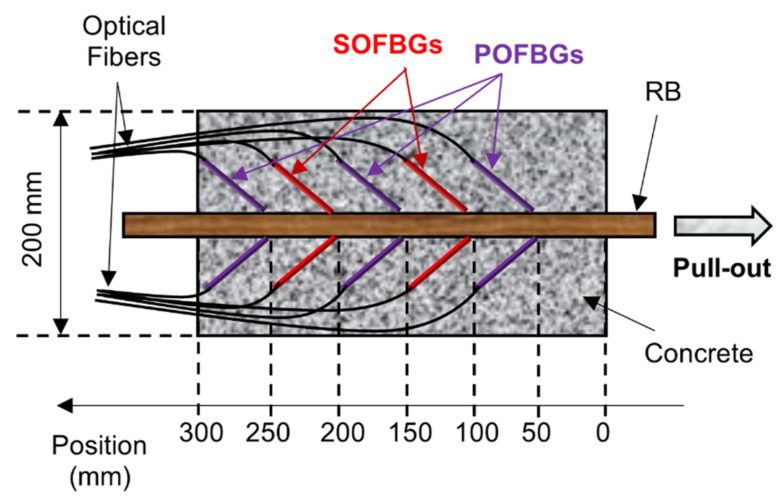
Location of the FBG based sensors inside the RC sample.

**Figure 4 sensors-22-08866-f004:**
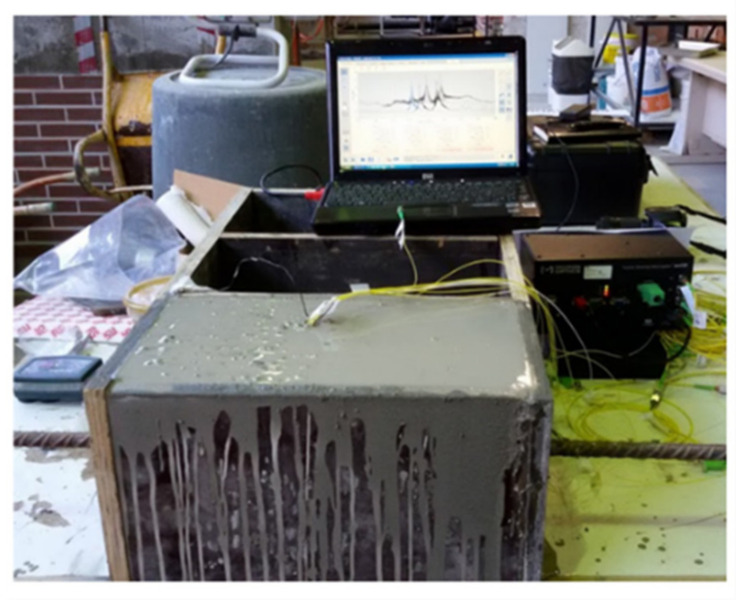
RC sample after filling with concrete.

**Figure 5 sensors-22-08866-f005:**
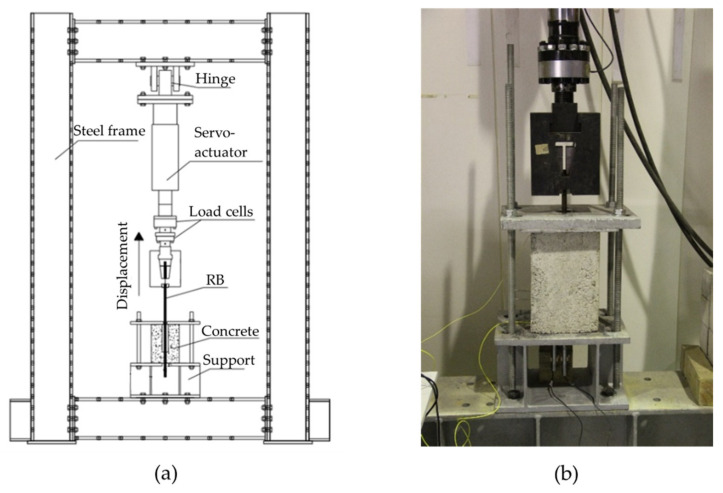
Pull-out testing setup: (**a**) Illustrative description. (**b**) Photography.

**Figure 6 sensors-22-08866-f006:**
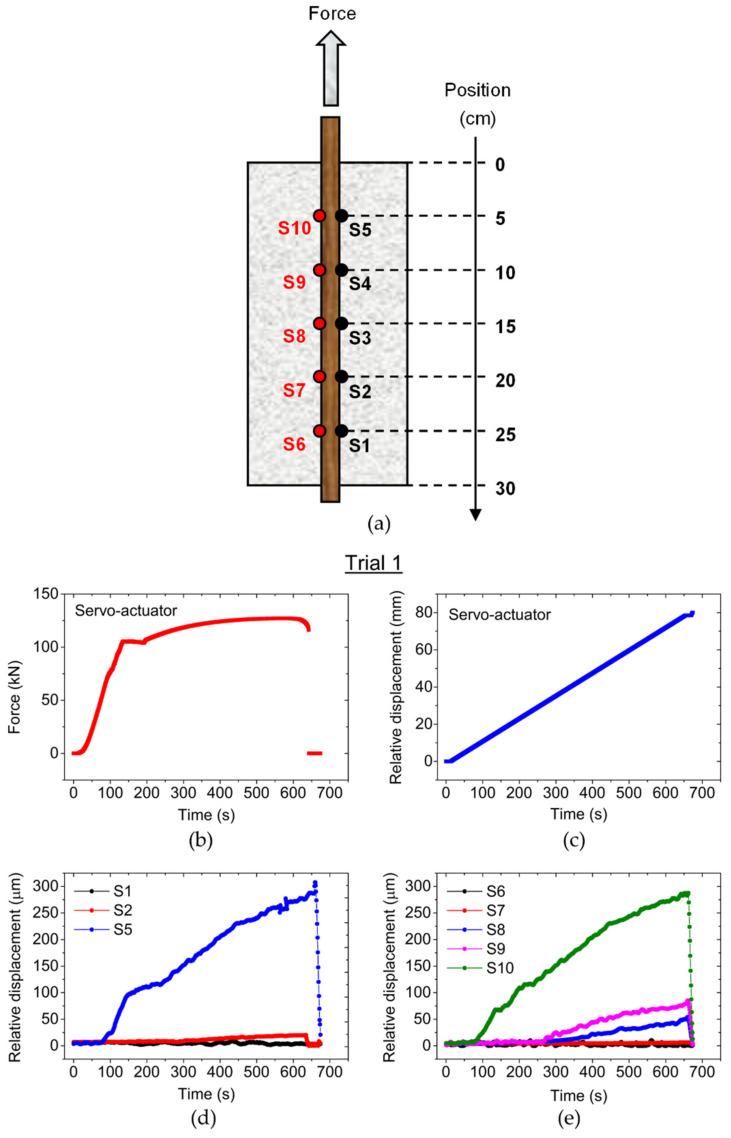
(**a**) Location of the optical sensors on RC sample. (**b**) Applied force to the RB during Trial 1. (**c**) Relative displacement from the servo-actuator. (**d**,**e**) Relative displacement measured by the optical sensors during Trial 1.

**Figure 7 sensors-22-08866-f007:**
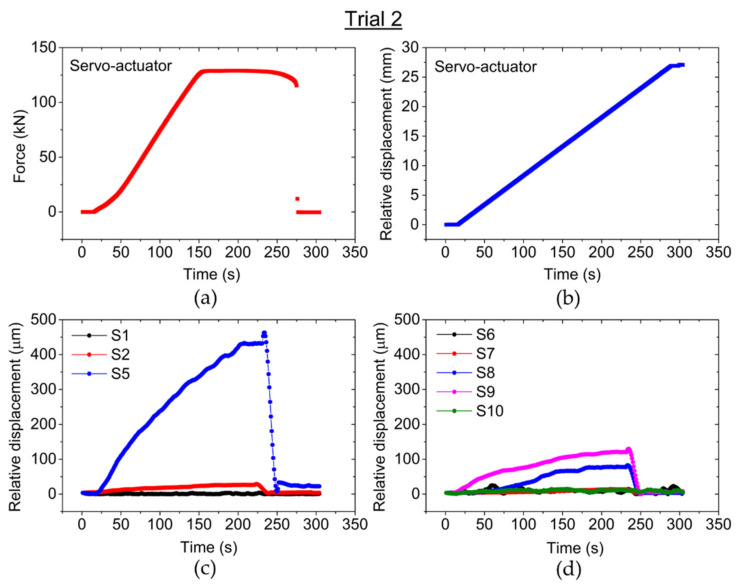
Experimental results from Trial 2: (**a**) Applied force to the RB during the pull-out test. (**b**) Relative displacement from the servo-actuator. (**c**,**d**) Relative displacement measured by the optical sensors during Trial 2.

**Figure 8 sensors-22-08866-f008:**
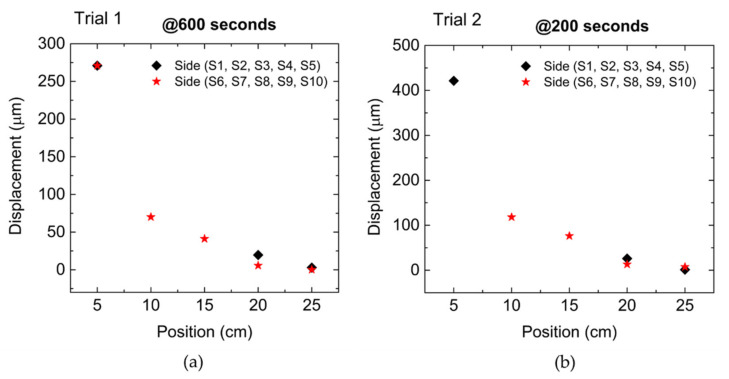
RB displacement measured by each FBG sensor: (**a**) At 600 s in Trial 1. (**b**) At 200 s in Trial 2.

**Figure 9 sensors-22-08866-f009:**
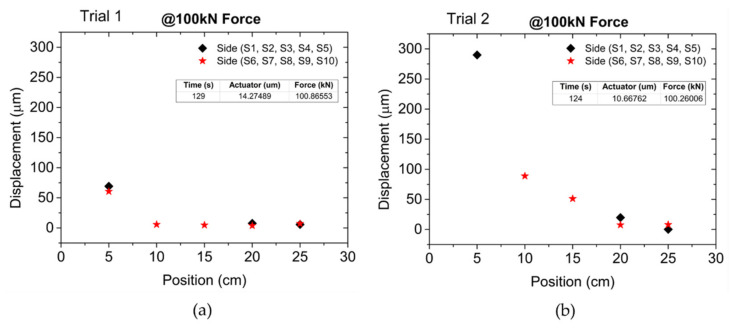
RB displacement measured by each sensor at 100 kN of pull-out force: (**a**) In Trial 1. (**b**) In Trial 2.

## Data Availability

The data presented in this study are available on request from the corresponding author.

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
