# Peer review of "Fiber Bragg Grating Sensors for Reinforcing Bar Slippage Detection and Bond-Slip Gradient Characterization"

_sensors, 2022, doi:10.3390/s22228866_

Round 1
Reviewer 1 Report
1.The work should be complemented with newer publications and studies from the years 2020 to 2022 on the use of optical sensors in the testing of reinforced concrete structures, relating in particular to: the adhesion and bond-slip of reinforcing steel in concrete.
2. Testing only one sample does not guarantee the reliability of the results obtained. Therefore, conclusions should only be limited to an assessment and comparison of the optical sensors used, i.e: POFBG and SOFBG and the effectiveness of used pull-out testing setup.
Author Response
Reviewer: 1
Comments and Suggestions for Authors:
- The work should be complemented with newer publications and studies from the years 2020 to 2022 on the use of optical sensors in the testing of reinforced concrete structures, relating in particular to: the adhesion and bond-slip of reinforcing steel in concrete.
Answer and changes included: Thank you for your comments. We agree with the Reviewer and we added the following information and references to the manuscript about the application of FBG sensors in reinforcing concrete structures:
“Čápová et al [14] installed sensors containing FBGs in a concrete bridge, by attaching those sensors to the reinforcing steel to monitor the prestressing of the structure in the initial phase and later the overall load of the structures, the impacts of temperature changes and traffic intensity. Kaklauskas et al [15] performed experimental and numerical studies to analyse the strain distribution on RC short tensile specimens, by installing in the RB tensor strain gauges and FBG sensors. Although the recordings from the FBGs were not as accurate as the strain gauges, the study highlighted the advantages of the FBG sensors in this type of application, namely the simple installation process, non-intrusive characteristics, the capability of using FBG sensors in smaller bar dimeters and the possibility of employing larger number of sensors with finer spacing between them. Other approach to evaluate the adherence zone between reinforcing bar and the concrete, and the durability of the structure is to detect and monitor corrosion, using FBGs to measure strain, temperature and refractive index in the surroundings of the RB [16]. In the case of corrosion monitoring based on strain measurements, the FBGs sensors are installed in the reinforcing bar to detect strain variations from the corrosion-induced expansion, as the corrosion products accumulate as result of the corrosion process [17–19].”
[14] Čápová, K.; Velebil, L.; Včelák, J. Laboratory and in-situ testing of intergrated FBG sensors for SHM for concrete and timber structures. Sensors 2020, 20, 1661.
[15] Kaklauskas, G.; Sokolov; A.; Ramanauskas, R.; Jakubovskis, R. Reinforcing strains in reinforcing concrete tensile members recorded by strain gauges and FBG sensors: experimental and numerical analysis. Sensors 2019, 19, 200.
[16] Fan, L.; Bao, Y. Review of fiber optic sensors for corrosion monitoring in reinforced concrete. Cem. Concr. Compos. 2021, 120, 104029.
[17] Mao, J.; Xu, F.; Gao, Q.; Liu, S.; Jin, W.; Xu, Y. A monitoring method based on FBG for concrete corrosion cracking. Sensors 2016, 16, 1093.
[18] Almubaied, O.; Chai, H. K.; Islam, M. R.; Lim, K.-S.; Tan, C. G. Monitoring Corrosion Process of Reinforced Concrete Structure Using FBG Strain Sensor. IEEE Trans. Instrum. Meas. 2017, 66, 2148–2155.
[19] Jamal, L.; Ahmad, S.; Qureshi, K. K. (2022) Reinforcement corrosion detection by using fiber Bragg grating based sensors. In Proceedings of the Conference on Lasers and Electro-Optics, San Jose, CA, USA, 15–20 May 2022.
- Testing only one sample does not guarantee the reliability of the results obtained. Therefore, conclusions should only be limited to an assessment and comparison of the optical sensors used, i.e: POFBG and SOFBG and the effectiveness of used pull-out testing setup.
Answer and changes included: Thank you for your comment. We agree with the Reviewer and besides the current information presented in the conclusions, regarding the capabilities and potential of this sensing configuration, we add the following information about the POFBG and SOFBG sensors and the need to perform more tests to validate and improve the current configuration:
“But these similarities are the result of the nature of the pull-out test. Despite the similar sensing capabilities of the POFBG and SOFBG devices, POFs are much more elastic than silica fiber and have higher strain limits, which allows to monitor higher displacement values. However, as the entire adherence zone collapsed during Trial 2, the tension accumulated in RB was released and it´s sudden displacement surpassed the strain limits of the sensors (SOFBGs and POFBGs), hence it was not possible to verify the elastic capabilities of the POFBGs.”
And,
“Also, despite the promising results, more tests are needed to confirm the viability of this sensing configuration, where the limitations of POF and silica fiber sensors are studied more intensively, the location and space between sensors are analysed, and the parameters of the pull-out test as well as the RC samples are altered to analyse the sensors performance under different bond-slip conditions.”

Reviewer 2 Report
In the manuscript presented by the authors, a configuration using ten optical fiber sensors is developed, allowing to monitor different sections of the same reinforced concrete (RC) structure. They note that since the reinforcing bar (RB) may suffer different strains along its length, the location of the sensors is critical, to provide an early warning about any displacement. The authors describe how the Bragg gratings were inscribed in both silica and polymer optical fibers and these devices worked as displacement sensors by monitoring the strain variations on the fibers. I agree that the results allowed to conclude that these sensors can be easily implemented in civil construction environment, and due to the small dimensions, they can be a non-intrusive technique when multiple sensors are implemented in the same RC structure. But I have to mentions a number of small revisions that should be done:
- In lines 156-161 you give info that you have used a random grating from the each group to test their sensitivities, but I am not sure that you could guarantee that all these gratings within the group are very similar. Particularly if you use non-commercial, self-manufacturing gratings. Please check their test data and then add the verified information to the text.
- The Conclusion section lacks technical data. I think some significant numbers given in Results and Discussion section should be also copied there.
- Some of the paper sections end with the figures, I propose to place them in the text, after the first mentioning.
Author Response
Comments and Suggestions for Authors:
In the manuscript presented by the authors, a configuration using ten optical fiber sensors is developed, allowing to monitor different sections of the same reinforced concrete (RC) structure. They note that since the reinforcing bar (RB) may suffer different strains along its length, the location of the sensors is critical, to provide an early warning about any displacement. The authors describe how the Bragg gratings were inscribed in both silica and polymer optical fibers and these devices worked as displacement sensors by monitoring the strain variations on the fibers. I agree that the results allowed to conclude that these sensors can be easily implemented in civil construction environment, and due to the small dimensions, they can be a non-intrusive technique when multiple sensors are implemented in the same RC structure. But I have to mentions a number of small revisions that should be done:
- In lines 156-161 you give info that you have used a random grating from the each group to test their sensitivities, but I am not sure that you could guarantee that all these gratings within the group are very similar. Particularly if you use noncommercial, self-manufacturing gratings. Please check their test data and then add the verified information to the text.
Answer and changes included: Thank you for your comments. Regarding the SOFBGs, it was used a commercial optical fiber (GF1 from Thorlabs) and the gratings were produced similarly in our laboratories. Although different gratings may present different sensing capabilities, the gratings used in this work have similar sensitivities, because all gratings are uniform FBGs, manufactured in the same optical fiber, and produced using the same inscription method (phase mask method) with the same parameters: same inscription laser system, same laser energy, same pulse frequency and same inscription time. The difference between gratings is the Bragg wavelength, but although it may influence the grating sensitivity, since they are all in the same spectral region, separated from each other by a few nanometers, the difference between sensitivities is very residual and insignificant, especially since the errors of the equipment used to apply strain on the fiber and to monitor the wavelength variation (plus the experimental errors) do not allow to differentiate the sensitivities of these gratings. In addition, the obtained strain sensitivity for the SOFBG (1.210 ± 0.003 pm/με) is in accordance with the typical value found in other GF1 fiber and standard single-mode optical fiber (1.2 pm/με) [1,2]. Regarding the POFBGs, the situation is the same except the fact that the fiber is non-commercial, and very small differences in fiber diameter may be found in these fibers, which may also affect the strain sensitivity of the gratings. However, the POF used to produce the POFBGs is same and any change in fiber diameter is very small and will have residual impact in the grating sensitivity, hence, like in the SOFBGs, sensitivity differences between gratings are most likely undetected due to the equipment and experimental errors. Also, the obtained sensitivity is relative close to the value found in [3], 1.46 pm/με, in which it was employed a similar step index single-mode poly(methyl methacrylate) (PMMA) POF with a slightly larger diameter (133 μm). In addition, other factors have higher impact in the displacement measures, like the possible relaxation of the fibre during the installation process and the replication of identical bond-slip sensors. Also, this work is a proof of concept, focused on the capability and viability of the sensing configuration to detect RB displacement in different section of the RC specimen.
Some information regarding the comparison of the obtained strain sensitivities values to the ones found the literature was added to the text.
- The Conclusion section lacks technical data. I think some significant numbers given in Results and Discussion section should be also copied there.
Answer and changes included: Thank you for your comments. We agree with the Reviewer and we added the following information to the Conclusions:
“In Trial 1, just before the interruption of the pull-out test, the maximum displacement was measured by the sensors located closer to the external force, S5 and S10, and the obtained values were 305.32 µm and 288.93 µm, respectively. During Trial 2, despite the failure of sensor S10 in measuring the movement of the RB, the sensor S5 was able to register a maximum displacement of 462.66 µm just before the pull out of the RB. On the other hand, the last section of the RC sample to be affected by the pull-out was the zone located further from the external force where the sensors S1 and S6 were installed, that measured maximum displacements of 3.70 µm and 8.53 µm, respectively, during Trial 2. Also, since Trial 2 is the repetition of the unfinished Trial 1, the initial conditions of the adherence zone between the RB and the concrete were different due to the external force applied to the bar during Trial 1 and consequently the optical sensors started measuring sooner the RB displacement during Trial 2 (at approximately 20 seconds, while in Trial 1 was at approximately 80 seconds). This can also explain the discrepancy between the maximum displacement values measured by the sensors in Trial 1 (69 µm) and in Trial 2 (290 µm), when the RB was subjected to the same applied force (100 kN).”
- Some of the paper sections end with the figures, I propose to place them in the text, after the first mentioning.
Answer and changes included: Thank you for your comments. We agree with the Reviewer and we rearranged the position of Figures 2, 4, 6, 7, 8 and 9.
REFERENCES
[1] Mesquita, E.; Pereira, L.; Theodosiou, A.; Alberto, N.; Melo, J.; Marques, C.; Kalli, K.; André, P.; Varum, H.; Antunes, P. Optical sensors for bond-slip characterization and monitoring of RC structures. Sensors Actuators A: Phys. 2018, 280, 332–339.
[2] Campanella, C. E.; Cuccovillo, A.; Campanella, C.; Yurt, A.; Passaro, V. M. N. Fibre Bragg grating based strain sensors: review of technology and applications. Sensors 2018, 18, 3115.
[3] Liu, H. Y.; Liu, H. B.; Peng, G. D. Tensile strain characterization of polymer optical fibre Bragg gratings. Opt. Commun. 2005, 251, 37–43.
